# The Prevalence and Risk Factors of Postpartum Depression among Mothers in Najran City, Saudi Arabia

**DOI:** 10.3390/healthcare12100986

**Published:** 2024-05-10

**Authors:** Majed Alshahrani, Nisreen Oudah Tami Alqarni, Sarah Saeed Aldughar, Shuruq Talea Asiri, Ruba Ibrahim Alharbi

**Affiliations:** 1Department of Obstetrics and Gynecology, College of Medicine, Najran University, Najran 61441, Saudi Arabia; 2College of Medicine, Najran University, Najran 11001, Saudi Arabia; nisreenalgrni509@gmail.com (N.O.T.A.); sarahseid426@gmail.com (S.S.A.); shroogasri8@gmail.com (S.T.A.); 3Batterjee Medical College, Jeddah 21442, Saudi Arabia; dr.rubaalharbi@gmail.com

**Keywords:** postpartum depression, mothers, Najran City, Saudi Arabia, prevalence, risk factors, EPDS, cross-sectional study

## Abstract

Background: Postpartum depression (PPD) is a significant mental health concern affecting mothers globally. However, research on PPD prevalence and risk factors in Najran City, Saudi Arabia, is limited. Study Aim: this cross-sectional study aimed to determine the prevalence and risk factors associated with PPD among mothers in Najran City. Methodology: A questionnaire-based study was conducted from September 2023 to January 2024, involving 420 mothers aged 16–50 years with newborns (2–10 weeks after delivery). The questionnaire included demographic information and the Arabic version of the Edinburgh Postnatal Depression Scale (EPDS). Statistical analysis utilized SPSS software v. 26, including descriptive statistics, Mann–Whitney *U* test, Kruskal–Wallis H test, and logistic regression. Results: The majority of participants were aged 20–35 years (61.4%), Saudi nationals (87.6%), and had university education (51.4%). EPDS scores indicated that 66.7% of mothers screened positive for possible depression. Significant associations were found between higher EPDS scores and factors such as unemployment (*p* = 0.004), younger age (*p* = 0.003), caesarean delivery (*p* = 0.043), mental illness (*p* = 0.0001), lack of adequate family support (*p* = 0.0001), and higher stress levels (*p* = 0.0001). Conclusion: The study revealed a high prevalence of PPD among mothers in Najran City, with sociodemographic, obstetric, and psychosocial factors significantly influencing PPD risk. These findings emphasize the need for targeted interventions and support systems to address maternal mental health needs effectively.

## 1. Background

Being a mother is a very challenging task, as it is a significant transitional period in which women encounter psychological, social, and physical changes that sometimes make women feel depressed, nervous, fearful, and confused [1,2].

Postpartum depression (PPD) is a significant depressive state that occurs during a period of 4–6 weeks after giving birth, and lasts for up to a year [3]. It is referred to as a pervasive complication of gestation, and is seen as a prevalent social and mental health issue [4]. It clarifies a variety of symptoms, such as feelings of insecurity, insomnia, anxiousness, lack of appetite, frustration, tearfulness, anger, low self-esteem, overwhelming sensations, and feeling guilty over not being able to care for their newborn [1].

Numerous risk factors have been identified that may make mothers more susceptible to developing postpartum depression. These include previous depressive episodes, a history of depression in the family, unemployment, low educational attainment, premature births, cesarean sections, unintended pregnancies, hormonal fluctuations during pregnancy, a lack of adequate husband and family support, marital conflicts, stressful life circumstances (such as being the victim of domestic violence), and traumatizing events [5]. In addition to that, the emergence of the COVID-19 outbreak made the condition much more difficult [6].

Postpartum depression is a public health concern [7] because it has been linked to serious consequences affecting the mother’s well-being and quality of life, making her emotionally unstable and unable to fulfill her role as a caregiver [8]. As a result of that, mother–infant bonding deteriorates, which leads to further complication regarding cognitive and psychological development of the child [9]. The issue may extend beyond that, and disrupts the harmony of the entire family [10].

Unfortunately, despite the high frequency of postpartum depression that has been shown in several studies, it is yet underdiagnosed, underlooked, undertreated, and most women do not even know about it [11]. Giving information on the prevalence of PPD could help to increase awareness, avoid its emergence in mothers who have known risk factors, and improve the processes of diagnosis, leading to more effective management [12]. Studies from various regions, including China, Ethiopia, Africa, and different cities within Saudi Arabia, have shed light on the multifactorial nature of PPD, emphasizing the importance of identifying risk factors to develop effective prevention and intervention strategies [13,14,15,16,17,18,19,20]. The present study aims to approximate the prevalence of postpartum depression and figure out potential risks related to PPD, in order to better understanding and develop preventative and management measures.

### 1.1. Study Aim

The aim of this study was to investigate the prevalence and risk factors of postpartum depression (PPD) among mothers in Najran City, Saudi Arabia, from September 2023 to January 2024.

### 1.2. Study Objectives

To determine the prevalence of postpartum depression among mothers in Najran City using the Edinburgh Postnatal Depression Scale (EPDS) questionnaire.

To identify demographic factors such as age, nationality, education, occupation, and health status associated with postpartum depression.To assess the impact of social support from family and stress levels on the occurrence of postpartum depression.To analyze the relationship between mode of birth (normal vaginal delivery vs. cesarean section) and the likelihood of experiencing postpartum depression among mothers in Najran City.

### 1.3. Methodology

A cross-sectional study design was employed to investigate the prevalence and risk factors of postpartum depression (PPD) among mothers in Najran City, Saudi Arabia. The study was conducted from September 2023 to January 2024, spanning a period of five months. This duration allowed for adequate recruitment of participants, data collection, and subsequent analysis, while considering seasonal variations that might impact maternal mental health. The research was conducted in Najran City, located in the southern region of Saudi Arabia. Najran City was chosen due to its diverse population and accessibility to healthcare facilities, providing a representative sample of postpartum mothers from different backgrounds and socioeconomic statuses.

The target population consisted of mothers aged between 16 and 50 years old who had recently given birth (2–10 weeks after delivery) and were residing in Najran City. Participants were included regardless of their nationality, occupation, or delivery method, ensuring a comprehensive representation of postpartum mothers in the region. Women who could not understand or fill out the form were excluded.

A non-probability convenience sampling technique was utilized to recruit participants for the study. This method involved selecting participants based on their accessibility and willingness to participate, which facilitated the recruitment process within the specified timeframe. The sample size calculation was based on an estimated prevalence rate of PPD and a confidence level of 95%.

The primary data collection tool used in this study was a structured questionnaire consisting of two main sections. The first section included demographic information such as age, nationality, education, occupation, health status, number of pregnancies, mode of birth, history of mental illness, family support, and stress levels. The second section comprised the Arabic version of the Edinburgh Postnatal Depression Scale (EPDS), a validated and reliable instrument used to assess maternal depressive symptoms during the postpartum period [21].

Regarding the EPDS score levels, it is essential to note that the scale ranges from 0 to 30, with higher scores indicating a higher level of depressive symptoms. The threshold commonly used to screen for PPD is a score of 12 or higher, which reflects a positive screen. However, it is important to understand the significance of other score ranges as well. Scores between 9 and 11 may indicate a moderate level of depressive symptoms, while scores between 12 and 13 may suggest the presence of possible depression. Scores of 14 and higher, as used in our study, are often considered indicative of a positive screen for PPD, warranting further evaluation and intervention.

Data collection was carried out using a self-administered questionnaire in Arabic, ensuring linguistic and cultural appropriateness for the participants. Trained research assistants distributed the questionnaires to eligible mothers, provided necessary instructions, and collected the completed questionnaires within a specified timeframe. Upon collection, the data were coded and entered into a secure electronic database to ensure confidentiality and accuracy. Data cleaning and validation procedures were conducted to identify and rectify any errors or inconsistencies in the dataset. The cleaned dataset was then subjected to statistical analysis using appropriate software, including descriptive statistics, chi-square tests, and logistic regression analysis to examine the prevalence and risk factors associated with PPD among the study participants. This comprehensive approach to data collection, management, and analysis enhances the validity and reliability of the study findings, and ensures reproducibility for future research in this field.

### 1.4. Ethical Considerations

Ethical approval for the study was obtained from the Institutional Review Board (IRB) of Najran University, Reference No: [012959-029281-DS], ensuring compliance with ethical guidelines and standards for human research. Informed consent was obtained from all participants before their inclusion in the study, emphasizing voluntary participation, confidentiality, and the right to withdraw at any stage without consequences. Measures were taken to safeguard the privacy and confidentiality of participants’ data throughout the study duration.

## 2. Results

Table 1 presents the characteristics of the 420 participants included in the study. The majority of participants were aged between 20 and 35 years (61.4%), followed by those above 35 years (31.9%), and a smaller proportion below 20 years (6.7%). Regarding nationality, the study predominantly consisted of Saudi mothers (87.6%) compared to non-Saudi mothers (12.4%). In terms of education, a significant portion of participants had university-level education (51.4%), while 39% had pre-university education, and 9.5% were illiterate. Most participants were not employed (64.3%), and the majority reported good health status (88.1%), with only a small percentage having comorbidities like gestational diabetes (6.7%), hypertension (3.3%), or thyroid disorders (1.9%).

Regarding obstetric history, the majority of participants were multiparous (71.9%) compared to nulliparous (28.1%). A substantial proportion of deliveries were via normal vaginal delivery (56.3%), with the remainder being cesarean deliveries (43.8%). Furthermore, about 43.8% of participants reported having had previous miscarriages, and 51.4% indicated that their pregnancies were planned. Mental illness was reported by 11% of participants, while 28.6% reported a lack of adequate family support. However, 71.4% of participants reported having support from their families, although only 63.3% were satisfied with the level of support. Additionally, 46.7% of participants reported experiencing psychological and social pressures, and 17.1% reported having stress levels that felt out of control. Concerning BMI, the majority fell within the normal range (18.5–24.9, 29%), while smaller proportions were underweight, overweight, or obese.

Table 2 displays the responses to the Edinburgh Postnatal Depression Scale (EPDS) items and score categories among the participants. The responses varied across the scale’s statements, reflecting diverse emotional states among postpartum mothers. For instance, a significant portion of mothers reported feeling less enjoyment (34.8%) and being unable to see the bright side of things (26.7%), while a considerable proportion reported feeling anxious or worried (47.1%) and having difficulty coping (39.5%). Moreover, a notable percentage reported feelings of sadness (32.9%) and unhappiness leading to crying (33.3%). However, fewer participants reported thoughts of self-harm (11.4%).

Regarding EPDS score levels, the majority of participants (66.7%) scored 14 or higher, indicating a positive screen for possible depression. Only a small percentage of participants scored less than 8 (14.3%), while 11.9% scored between 9 and 11, and 7.1% scored between 12 and 13 (Figure 1).

The provided data are the results of various statistical analyses, including Mann–Whitney *U* tests and Kruskal–Wallis H tests, conducted to assess the relationship between different variables and the EPDS scores, which measure the severity of postnatal depression symptoms (Table 3). There is no significant difference in EPDS scores between Saudi and non-Saudi individuals (*p* = 0.269). Individuals without a job (unemployed) have significantly higher EPDS scores compared to those with a job (*p* = 0.004). Multiparous individuals (those with multiple pregnancies) have lower EPDS scores compared to nulliparous individuals (those with not given birth) (*p* = 0.026). Individuals who underwent cesarean delivery have significantly higher EPDS scores compared to those who had a normal vaginal delivery (*p* = 0.043). There is no statistically significant difference in EPDS scores between individuals with and without a history of previous miscarriages (*p* = 0.065). Individuals with unplanned pregnancies have higher EPDS scores compared to those with planned pregnancies, although the difference is marginally significant (*p* = 0.05). Individuals suffering from mental illness have significantly higher EPDS scores compared to those without mental illness (*p* < 0.0001). The lack of adequate support from the family is associated with significantly higher EPDS scores compared to those with family support (*p* < 0.0001). Individuals who are not satisfied with the level of support from their family have significantly higher EPDS scores compared to those who are satisfied (*p* < 0.0001). Individuals experiencing psychological and social pressures have significantly higher EPDS scores compared to those who do not (*p* < 0.0001). Younger individuals (less than 20 years old) have significantly higher EPDS scores compared to those aged 20–35 and those older than 35 (*p* = 0.004). There is no significant difference in EPDS scores among individuals with different levels of education (*p* = 0.205). Individuals with diabetes (GD), hypertension (HTN), and thyroid issues have significantly higher EPDS scores compared to those without these health conditions (*p* = 0.001). EPDS scores increase significantly with the level of stress, with those reporting stress as “out of control” having the highest scores (*p* < 0.0001). There is no significant difference in EPDS scores among individuals with different BMI levels (*p* = 0.780).

Overall, these results highlight various factors associated with higher EPDS scores, indicating an increased risk or severity of postnatal depression symptoms. These factors include unemployment, cesarean delivery, unplanned pregnancy, mental illness, lack of adequate family support, dissatisfaction with family support, psychological and social pressures, younger age, certain health conditions, and high levels of stress.

A multiple linear regression was run to predict EPDS score from different predictors (Table 4). These variables statistically significantly predicted EPDS score, F (15, 400) = 9.314, *p* < 0.0005, R^2^ = 0.259. All variables added statistically significantly to the prediction, *p* < 0.05. For each one unit increase in age, the odds of depression decrease by approximately 1.471 times (with a 95% confidence interval from −2.427 to −0.515), holding all other variables constant. This effect is statistically significant at *p* = 0.003. There is no significant association between nationality and depression, as the odds ratio (1.020) is not statistically significant (*p* = 0.228). Higher levels of education are associated with slightly lower odds of depression, but this effect is not statistically significant (*p* = 0.088). Each one unit increase in job status is associated with a decrease in the odds of depression by approximately 1.627 times (*p* = 0.005). Health status does not significantly influence the odds of depression (*p* = 0.157). Body Mass Index (BMI) also does not significantly affect the odds of depression (*p* = 0.421). There is no statistically significant association between the number of pregnancies and depression (*p* = 0.081). The odds of depression increase by approximately 1.313 times if there has been a birth (*p* = 0.020). Previous miscarriages do not significantly affect the odds of depression (*p* = 0.101). Unplanned pregnancies are associated with lower odds of depression by approximately 1.103 times (*p* = 0.048). Individuals suffering from mental illness have much higher odds (approximately 3.520 times) of depression (*p* < 0.0001). A lack of adequate family support is associated with lower odds (approximately 2.073 times) of depression (*p* = 0.001). Decreased satisfaction with family support is associated with lower odds (approximately 3.110 times) of depression (*p* < 0.0001). Individuals experiencing psychological and social pressures have significantly higher odds (approximately 4.126 times) of depression (*p* < 0.0001). Higher levels of stress are associated with higher odds (approximately 2.805 times) of depression (*p* < 0.0001).

## 3. Discussion

Postpartum depression (PPD) is a prevalent mental health issue among new mothers globally, characterized by feelings of sadness, hopelessness, and anxiety that can significantly impact maternal and infant well-being [1,2,3]. Our study focused on Najran City, Saudi Arabia, and revealed a concerning prevalence of PPD, with 66.7% of mothers screening positive for possible depression based on the EPDS scores. This finding aligns with similar studies conducted in other regions of Saudi Arabia, such as Jazan, Riyadh, Al-Madina, Jeddah, and the Qassim Region, where PPD prevalence rates ranged from 19.4% to 75.7%, indicating a significant mental health burden among postpartum women across the country [16,17,18,19,20].

Studies highlighted several risk factors associated with PPD, including gestational diabetes, depression during pregnancy, history of depression, poor social support, lower economic status, and adverse obstetric outcomes, among others [13,14,15]. In our study, we observed significant associations between higher EPDS scores and factors such as unemployment, younger age, cesarean delivery, mental illness, lack of adequate family support, and elevated stress levels. These findings corroborate with the existing literature, emphasizing the consistent impact of psychosocial, obstetric, and demographic factors on PPD risk [16,17,18,19,20].

For instance, a study in Jazan emphasized the pivotal role of family support, highlighting a 5.9-fold increased risk of PPD among mothers lacking adequate family support, which echoes our findings regarding the significant association between low family support and higher EPDS scores [16]. Similarly, studies in Riyadh and the Qassim Region emphasized the influence of unsupportive spouses, stressful life events, and social support on PPD risk, reinforcing the importance of social and familial support systems in mitigating PPD [17,20].

Furthermore, our study noted a higher risk of PPD among primiparous women and those delivering via the normal vaginal method compared to multiparous women and cesarean deliveries, respectively, aligning with findings from Al-Madina, which reported a 1.91-fold increased risk among primiparous women and a 3.11-fold increased risk among those with vaginal deliveries [18]. These findings suggest the need for tailored interventions and support mechanisms for women based on their obstetric history and delivery method.

One of the most significant findings of our study was the strong association between mental illness history and PPD risk (OR = 3.520, *p* < 0.0001). This underscores the critical need for comprehensive mental health screening and support for pregnant and postpartum women with a history of mental health disorders, consistent with recommendations from global health organizations [21]. Integrating mental health services into routine antenatal and postnatal care can help identify at-risk individuals early and provide timely interventions.

Additionally, our study highlighted the detrimental impact of inadequate family support on maternal mental health, as evidenced by significantly higher EPDS scores among mothers lacking adequate family support (*p* < 0.0001). Strengthening family-centered support programs and promoting open communication within families can contribute significantly to reducing the PPD burden and enhancing maternal well-being. It is important to consider additional initiatives during pregnancy and in the community to complement the postpartum assessment. Future studies or programs aimed at addressing postpartum depression could benefit from incorporating pre-pregnancy and pregnancy assessments. These initiatives contribute to the development of more comprehensive and effective interventions for maternal mental health, thereby improving overall maternal well-being.

## 4. Conclusions

In conclusion, our study underscores the high prevalence of PPD among mothers in Najran City, Saudi Arabia, and emphasizes the multifaceted nature of PPD risk factors, including sociodemographic, obstetric, and psychosocial determinants. Integrating mental health screening, enhancing family support systems, and tailoring interventions based on individual risk profiles are crucial steps in addressing PPD and promoting maternal mental health and well-being.

## Figures and Tables

**Figure 1 healthcare-12-00986-f001:**
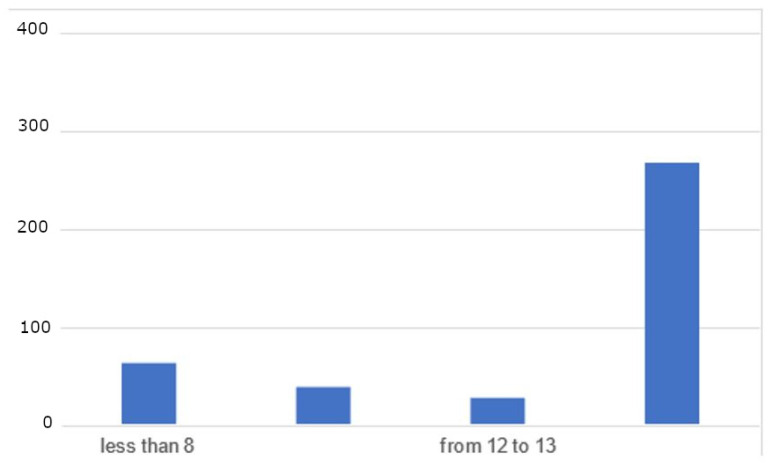
EPDS score categories distribution.

**Table 1 healthcare-12-00986-t001:** Characters of the included participants (*n* = 420).

Parameter		Frequency (%)
Age	Less than 20	28 (6.7%)
	20–35	258 (61.4%)
	More than 35	134 (31.9%)
Nationality	Saudi	368 (87.6%)
	Non-Saudi	52 (12.4%)
Education	Illiterate	40 (9.5%)
	Pre-university	164 (39%)
	University	216 (51.4%)
Job	No	270 (64.3%)
	Yes	150 (35.7%)
Health status	No	370 (88.1%)
	DM	28 (6.7%)
	HTN	14 (3.3%)
	THYROID	8 (1.9%)
How many pregnancies?	Nulli-para	118 (28.1%)
	Multi-para	302 (71.9%)
Mode of Birth	Normal virginal delivery	234 (56.3%)
	Caesarean	182 (43.8%)
Have there been any previous miscarriages?	No	236 (56.2%)
	Yes	184 (43.8%)
Was the pregnancy planned?	No	204 (48.6%)
	Yes	216 (51.4%)
Do you suffer from mental illness?	No	374 (89%)
	Yes	46 (11%)
Is there support from the family?	No	120 (28.6%)
	Yes	300 (71.4%)
Are you satisfied with the level of support from your family?	No	154 (36.7%)
	Yes	266 (63.3%)
Do you suffer from psychological and social pressures?	No	224 (53.3%)
	Yes	196 (46.7%)
What is the level of stress?	Nothing	180 (42.9%)
	Under control	168 (40%)
	Out of control	72 (17.1%)
BMI category	less than 18.5	14 (3.3%)
	18.5–24.9	122 (29%)
	25–29.9	166 (39.5%)
	30–34.9	80 (19%)
	35–39.9	20 (4.8%)
	more than 40	18 (4.3%)

**Table 2 healthcare-12-00986-t002:** EPDS item responses and score categories (*n* = 420).

Parameter		Frequency (%)
I have looked forward with enjoyment to things	As much as I ever did	146 (34.8%)
	Rather less than I used to	140 (33.3%)
	Definitely less than I used to	112 (26.7%)
	Hardly at all	22 (5.2%)
I have been able to laugh and see the bright side of things?	As much as I ever did	154 (36.7%)
	Rather less than I used to	156 (37.1%)
	Definitely less than I used to	92 (21.9%)
	Hardly at all	18 (4.3%)
I have blamed myself unnecessarily when things went wrong	No, never	68 (16.2%)
	Not very often	112 (26.7%)
	Yes, some of the time	128 (30.5%)
	Yes, most of the time	112 (26.7%)
I have been anxious or worried for no good reason	No, not at all	44 (10.5%)
	Hardly ever	78 (18.6%)
	Yes, sometimes	100 (23.8%)
	Yes, very often	198 (47.1%)
I have felt scared or panicky for no very good reason	No, not much	106 (25.2%)
	Yes, sometimes	170 (40.5%)
	Yes, quite a lot	144 (34.3%)
Things have been getting on top of me	No, I have been coping as well as ever.	44 (10.5%)
	No, most of the time I have coped quite well.	70 (16.7%)
	Yes, sometimes I haven’t been coping as well as usual	140 (33.3%)
	Yes, most of the time I haven’t been able to cope at all.	166 (39.5%)
I have been so unhappy that I have had difficulty sleeping	No, not at all	36 (8.6%)
	Not very often	110 (26.2%)
	Yes, sometimes	114 (27.1%)
	Yes, most of the time	160 (38.1%)
I have felt sad or miserable	No, not at all	66 (15.7%)
	Not very often	124 (29.5%)
	Yes, quite often	138 (32.9%)
	Yes, most of the time	92 (21.9%)
I have been so unhappy that I have been crying	No, never	64 (15.2%)
	Only occasional	110 (26.2%)
	Yes, quite often	106 (25.2%)
	Yes, most of the time	140 (33.3%)
The thought of harming myself has occurred to me	Never	236 (56.2%)
	Hardly ever	78 (18.6%)
	Sometimes	58 (13.8%)
	Yes, quite often	48 (11.4%)
EPDS score level	less than 8	60 (14.3%)
	9–11	50 (11.9%)
	12–13	30 (7.1%)
	14 and higher and positive screen	280 (66.7%)

**Table 3 healthcare-12-00986-t003:** EPDS scores in association with the characters of respondents (*n* = 420).

Variables	N	EPDS Score (Mean ± SD)	*p*-Value
**Nationality**			
Saudi	368	14.83 ± 5.673	*p*-value *=* 0.269 *U* * = 8664
Non-Saudi	52	15.85 ± 5.932	
**Job**			
No	270	15.53 ± 5.560	*p*-value *=* 0.004 *U* ** = 16,796
Yes	150	13.91 ± 5.842	
**How many pregnancies?**			
Nulli-para	118	15.73 ± 5.832	*p*-value *=* 0.026 *U* * = 15,328
Multi-para	302	14.65 ± 5.641	
**Birth**			
Normal virginal delivery	234	14.40 ± 5.767	*p*-value *=* 0.043 *U* * = 18,840
Cesarean	182	15.71 ± 5.610	
**Have there been any previous miscarriages?**			
No	236	15.36 ± 5.889	*p*-value *=* 0.065 *U* * = 19,438
Yes	184	14.43 ± 5.441	
**Was the pregnancy planned?**			
No	204	15.52 ± 6.070	*p*-value *=* 0.05 *U* * = 19,610
Yes	216	14.42 ± 5.304	
**Do you suffer from mental illness?**			
No	374	14.57 ± 5.765	*p*-value *=* 0.0001 *U* * = 5210
Yes	46	18.09 ± 4.087	
**Is there support from the family?**			
No	120	16.43 ± 5.434	*p*-value *=* 0.0001 *U* * = 13,988
Yes	300	14.36 ± 5.717	
**Are you satisfied with the level of support from your family?**			
No	154	16.92 ± 5.345	*p*-value *=* 0.0001 *U* * = 13,752
Yes	266	13.81 ± 5.610	
**Do you suffer from psychological and social pressures?**			
No	224	13.03 ± 5.466	*p*-value *=* 0.0001 *U* * = 12,974
Yes	196	17.15 ± 5.171	
**Age**			
Less than 20	28	17.43 ± 3.024	*p*-value *=* 0.004 χ^2^ ** = 11.040
20–35	258	15.19 ± 5.766	
More than 35	134	13.99 ± 5.852	
**Education**			
Illiterate	40	15.70 ± 6.009	*p*-value *=* 0.205 χ^2^ ** = 3.172
Pre-university	164	15.39 ± 5.509	
University	216	14.48 ± 5.786	
**Health status**			
No	370	14.72 ± 5.742	*p*-value *=* 0.001 χ^2^ ** = 16.403
Dm	28	16.71 ± 5.241	
Htn	14	19.57 ± 4.108	
Thyroid	8	11.50 ± 1.927	
**What is the level of stress?**			
Nothing	180	12.94 ± 5.303	*p*-value *=* 0.0001 χ^2^ ** = 52.634
Under control	168	15.51 ± 5.440	
Out of control	72	18.67 ± 5.184	
**EPDS score level**			
Less than 8	60	5.27 ± 1.561	*p*-value *=* 0.0001 χ^2^ ** = 293.926
9–11	50	9.84 ± 1.201	
12–13	30	12.40 ± 0.498	
14 and higher and positive screen	280	18.21 ± 3.451	
**BMI Category**			
Less than 18.5	14	14.57 ± 6.699	*p*-value *=* 0.780 χ^2^ ** = 2.478
18.5–24.9	122	15.49 ± 5.502	
25–29.9	166	14.95 ± 5.531	
30–34.9	80	14.68 ± 6.696	
35–39.9	20	13.70 ± 3.213	
More than 40	18	14.22 ± 5.631	

*p*-value < 0.05 is statistically significant; * Mann–Whitney *U* test ** Kruskal–Wallis H test.

**Table 4 healthcare-12-00986-t004:** Predictors of postnatal depression among participants.

Variables	Odds Ratio	95% *CI* *		*p*-Value *
		Lower	Upper	
Constant	16.628	11.459	21.796	0.0001
Age	−1.471	−2.427	−0.515	0.003
Nationality	1.020	−0.641	2.682	0.228
Education	−0.722	−1.552	0.107	0.088
Job	−1.627	−2.760	−0.493	0.005
Health status	0.681	−0.264	1.625	0.157
BMI	−0.033	−0.115	0.048	0.421
How many pregnancies?	−1.080	−2.295	0.135	0.081
Birth	1.313	0.205	2.420	0.020
Have there been any previous abortion?	−0.921	−2.023	0.180	0.101
Was the pregnancy planned?	−1.103	−2.195	−0.011	0.048
Do you suffer from mental illness?	3.520	1.798	5.243	0.0001
Is there support from the family?	−2.073	−3.270	−0.876	0.001
Are you satisfied with the level of support from your family?	−3.110	−4.208	−2.012	0.0001
Do you suffer from psychological and social pressures?	4.126	3.102	5.151	0.0001
What is the level of stress?	2.805	2.104	3.505	0.0001

* 95% *CI*: confidence interval and *p*-value: significant if *p* ≤ 0.05 and non-significant if *p* ≥ 0.05.

## Data Availability

Data are contained within the article.

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
