# Peer review of "The Prevalence and Risk Factors of Postpartum Depression among Mothers in Najran City, Saudi Arabia"

_healthcare, 2024, doi:10.3390/healthcare12100986_

Round 1
Reviewer 1 Report
Comments and Suggestions for Authors
I have read this paper with great interest, and value the effort and data as reported as this is another illustration of the relevance and prevalence of postpartum depression and mental issues in this specific setting. I do however have additional comments and suggestions to further improve the messages, and how readers should understand this work.
First, it is not clear to this reviewer when the questionnaire has been collected, or I have missed this information. I assume that this is (very) shortly after delivery. In my opinion, this may impact the prevalence findings, so that this should be added to the abstract, and be clearly mentioned in the methods section.
Second, I understand that the authors made efforts to have a representative cohort of respondents. It would however be valuable to compare the key findings (e.g. age, multi/primi, educational level) with population data on women who deliver in the region or the country.
Finally and although I do value the work provided, there is very likely more add on value to improve this by additional initiatives in the community, and during pregnancy, in addition to a postpartum assessment. In this way, these data do show the relevance, but methodologically, we need more links and information on pre-pregnancy and pregnancy to develop effective programs. That’s perhaps rather an opinion, but I do think that there is add on value to reflect somewhat on this in the discussion part of the paper.
Specific comments
Typo: virginal delivery (4x), so read vaginal delivery.
Line 127: 6.7 % diabetes mellitus ? likely gestational diabetes (as mentioned in the discussion)
Line 132 (and elsewhere): do you really mean abortus, or miscarriage ?
Line 159: no previous pregnancy ?
You have focused on the EDPS score level of 14, reflecting a positive screen. I would (cfr figure 1) also somewhat better explain the other ‘thresholds’ of the EDPS score.
Lack of support: is this not rather how it has been perceived by the mother, not necessary equal to the ‘real’ setting (in both directions). (Line 257 discussion, but also in the results section)
Author Response
Comment |
Action |
It is not clear to this reviewer when the questionnaire has been collected, or I have missed this information. I assume that this is (very) shortly after delivery. In my opinion, this may impact the prevalence findings, so that this should be added to the abstract, and be clearly mentioned in the methods section |
(2-10 weeks after delivery)
|
I do value the work provided, there is very likely more add on value to improve this by additional initiatives in the community, and during pregnancy, in addition to a postpartum assessment. In this way, these data do show the relevance, but methodologically, we need more links and information on pre-pregnancy and pregnancy to develop effective programs. That’s perhaps rather an opinion, but I do think that there is add on value to reflect somewhat on this in the discussion part of the paper |
Thank you for your comment. This is limitation of our study we don't include the antenatal assessment with postnatal assessment |
Typo: virginal delivery (4x), so read vaginal delivery.
|
Corrected |
Line 127: 6.7 % diabetes mellitus ? likely gestational diabetes (as mentioned in the discussion)
|
Corrected to gestational diabetes |
Line 132 (and elsewhere): do you really mean abortus, or miscarriage ?
|
Corrected to miscarriage |
Line 159: no previous pregnancy ? |
Its mean not giving birth) |
You have focused on the EDPS score level of 14, reflecting a positive screen. I would (cfr figure 1) also somewhat better explain the other ‘thresholds’ of the EDPS score. |
We focused her how get positive screening for EDPS |
Lack of support: is this not rather how it has been perceived by the mother, not necessary equal to the ‘real’ setting (in both directions). (Line 257 discussion, but also in the results section)
|
Yes , its mean how the mother feeling from family |

Reviewer 2 Report
Comments and Suggestions for Authors
Firstly, I would like to congratulate the authors for the research.
I would like to contribute some considerations regarding the manuscript.
Introduction: the first paragraph needs revision. The paragraph is too long and the information must be organized.
I would like to suggest that research related to the topic carried out with the population of Saudi Arabia be included in the text. These studies would support the theoretical framework, differentiate this study from others, and emphasize its importance.
Method: I suggest adding whether there were excluded participants and the exclusion criteria.
Characterize research protocols and procedures in more detail for reproducibility purposes.
Results: the writing of the results requires revision. The results are described in a combined way, making understanding difficult. I suggest separating by topic of analysis, including tables in the body of the text, closer to the explanation.
Discussion: I suggest that the authors consider their hypotheses regarding the findings and the city researched, as well as provide explanations and comparisons of the observations of the researchers of the present study on why this study is different from others in Saudi Arabia.
Author Response
Introduction: the first paragraph needs revision. The paragraph is too long and the information must be organized. |
Thank you for your comment but the introduction is describe the Postpartum depression , the risk factor , the concerning the public health regarding the Postpartum depression and aim of the study |
Method: I suggest adding whether there were excluded participants and the exclusion criteria |
Done in method section Participants were included regardless of their nationality, occupation, or delivery method, ensuring a comprehens- ive representation of postpartum mothers in the region. Women who could not understand or fill out the form were excluded |
Results: the writing of the results requires revision. The results are described in a combined way, making understanding difficult. I suggest separating by topic of analysis, including tables in the body of the text, closer to the explanation. |
Done with separating by topic of analysis, including tables in the body of the text, closer to the explanation |
Discussion: I suggest that the authors consider their hypotheses regarding the findings and the city researched, as well as provide explanations and comparisons of the observations of the researchers of the present study on why this study is different from others in Saudi Arabia.
|
Its explain in discussion the our finding with other study in Saudi Arabia such as Jazan, Riyadh, Al-Madina, Jeddah, and the Qassim Region |

Round 2
Reviewer 2 Report
Comments and Suggestions for Authors
I have no comments on the content of the text, I congratulate the authors for their efforts. There is just one observation: there are many words in the text that are divided by a dash.
Author Response
Thank you.
according to reviewer, there is no comments on the content of the text.